# Synthesis and Biological Evaluation of Cassane Diterpene (5α)-Vuacapane-8(14), 9(11)-Diene and of Some Related Compounds

**DOI:** 10.3390/molecules27175705

**Published:** 2022-09-04

**Authors:** Houda Zentar, Fatin Jannus, Marta Medina-O’Donnell, José A. Lupiáñez, José Justicia, Ramón Alvarez-Manzaneda, Fernando J. Reyes-Zurita, Enrique Alvarez-Manzaneda, Rachid Chahboun

**Affiliations:** 1Departamento de Química Orgánica, Facultad de Ciencias, Instituto de Biotecnología, Universidad de Granada, 18071 Granada, Spain; 2Departamento de Bioquímica y Biología Molecular I, Facultad de Ciencias, Universidad de Granada, 18071 Granada, Spain; 3Área de Química Orgánica, Departamento de Química y Física, Universidad de Almería, 04120 Almería, Spain

**Keywords:** natural products, semisynthesis, cassane diterpenoids, anti-inflammatory activity, anti-cancer activity

## Abstract

A set of thirteen cassane-type diterpenes was synthesized and an expedient synthetic route was used to evaluate 14-desmethyl analogs of the most active tested cassane. The anti-inflammatory activities of these 13 compounds were evaluated on a lipopolysaccharide (LPS)-activated RAW 264.7 cell line by inhibition of nitric oxide (NO) production, some of them reaching 100% NO inhibition after 72 h of treatment. The greatest anti-inflammatory effect was observed for compounds **16** and **20** with an IC_50 NO_ of 2.98 ± 0.04 μg/mL and 5.71 ± 0.14 μg/mL, respectively. Flow-cytometry analysis was used to determine the cell cycle distribution and showed that the inhibition in NO release was accompanied by a reversion of the differentiation processes. Moreover, the anti-cancer potential of these 13 compounds were evaluated in three tumor cell lines (B16-F10, HT29, and Hep G2). The strongest cytotoxic effect was achieved by salicylaldehyde **20**, and pterolobirin G (**6**), with IC_50_ values around 3 μg/mL in HT29 cells, with total apoptosis rates 80% at IC_80_ concentrations, producing a significant cell-cycle arrest in the G0/G1 phase, and a possible activation of the extrinsic apoptotic pathway. Additionally, initial *SAR* data analysis showed that the methyl group at the C-14 positions of cassane diterpenoids is not always important for their cytotoxic and anti-inflammatory activities.

## 1. Introduction

Cassane diterpenoids are a group of rearranged abietane metabolites isolated from different species of the *Caesalpinia* genus, where they are the predominant characteristic components of these plants. Up to now, more than 450 cassane diterpenoids have been discovered based only on phytochemical investigations of *Caesalpinia* species [1]. They have received much attention for their structural diversity, which in some cases is complex, for example, tricyclic cassane diterpenoids with a fused furan ring or butanolide lactone, tricyclic cassane diterpenoids, norcassane diterpenoids, and other types [2]. Natural cassane diterpenes exhibit a wide range of pharmacological activities, such as anti-inflammatory, antitumor, antimalarial, antimicrobial, antiviral, antioxidant, antiplasmodial, and antinociceptive properties [3,4]. Some of these compounds, bearing an aromatic C ring, have gained special attention due to their significant pharmaceutical potential. Researchers have reported only a few syntheses of these interesting diterpenoids due to their relatively complex structures. In this sense, Pitsinos et al. reported the synthesis of 14-desmethyl taepeenin D (**2**) and a series of second-generation analogs of taepeenin D (**1**) with the aim of investigating their structure–activity relationship (*SAR*) [5]. These authors argued that the presence of the methyl group in C-14 does not influence the anticancer activity of this type of compound [6]. Our group has achieved the first synthesis of taepeenin F (**3**) from dehydroabietic acid [7], as well as the first synthesis of the putative structure of benthaminin 1 (**4**) from *trans*-communic acid [8]. This cassane diterpene displays antibacterial activity, with an MIC value of 47.8 μM for both *Staphylococcus aureus* and *Micrococcus flavus* [9]. Recently, we also developed a novel process for the synthesis of pterolobirin H (**5**), pterolobirin G (**6**) and (5a)-vouacapane-8(14), 9(11)-diene (**7**), starting from (+)-sclareolide (**8**). The utilized synthetic strategy does not require the use of protecting groups, as it consists of only 10 steps with an overall yield of 20% and was developed with total atoms economy [10]. Wang et al. [11] reported that the cassane (5a)-vouacapane-8(14),9(11)-diene (**7**) exhibits anti-inflammatory properties by inhibiting the nitric oxide production ratio (34.5%) in the RAW 264.7 cells. However, only a few total syntheses of cassane type viridin (**9**) [12,13], sucutinirane C (**10**), and sucutinirane D (**11**) have been reported [14]. Viridin (**9**) was shown to have a strong fungistatic effect, blocking the germination of *Botrytis allii*, *Colletotrichum lini*, and *Fusarium caeruleum* spores at (0.003–0.006) μg/mL [15]. Furthermore, it has been demonstrated that sucutinirane C (**10**) displays good NO inhibition with an IC_50_ value of 24.44 ± 1.34 μM [16] (Figure 1).

Our main aim was to continue efforts into the search for bioactive compounds and to investigate more structurally novel diterpene cassane derivatives. In this study, we have performed biological assays of the unprecedented synthesized cassanes (5a)-vouacapane-8(14), 9(11)-diene (**7**), pterolobirins H (**5**) and G (**6**), and other selected intermediates [10]. In parallel, the other objective of this research is the novel synthesis of some 14-demethylated cassane analogs, structurally related to the most active compound to advance current knowledge on their structure-activity relationship (*SAR*) For this purpose, we used (-)-ferruginol (**12**) as a starting product, which is easily prepared from commercial (+)-dehydroabietylamine [17,18].

All the selected compounds were evaluated for the first time for their antiproliferative activity against HT-29, B16F10, and HepGe-2 cancer cell lines, as well as for their NO inhibitory activity against the lipopolysaccharide (LPS)-induced murine macrophage RAW-264.7cell line. Furthermore, the most active compounds were selected for the subsequent cytometry study on a selected cancer cell line with the lowest IC_50_ values, for a thorough biological investigation, which is also one of our purposes of this paper.

## 2. Results and Discussion

### 2.1. Chemistry

Based on our previous study on the synthesis of (5a)-vouacapane-8(14),9(11)-diene (**7**) and related compounds [10], we selected ten compounds (**5**–**7** and **14**–**20**) with a cassane skeleton for biological assays. All the selected compounds, were obtained from diene **13**, easily prepared from (+)-sclareolide (**8**) according to the process described by our research group [19,20]. Diels–Alder cycloaddition of diene **13** with DMAD afforded the phthalate derivative cycloadduct **14** when the reaction was carried out in toluene at reflux. Under similar conditions, but in a sealed tube, cycloadduct **15** was obtained and oxidized to dienone **16** with catalytic PDC in the presence of TBHP (Figure 1).

Compounds **17**–**20** were prepared from dienone **16.** After treatment with BF_3_.Oet_2_ at room temperature, **16** was converted directly to phenol ester **18** in high yield. However, when **16** was treated with the I_2_/PPh_3_ system, phenol diester **17** was unexpectedly obtained. This phenol is generated after the loss of methyl iodide, promoted by the protonation of the carbonyl group with iodic acid generated in the reaction medium. The further reduction in **18** with NaBH_4_ gave hydroxy phenol **19**, which re-oxidized selectively with activated MnO_2_ in DCM to produce hydroxy aldehyde **20**. Lactone **5** was obtained after the insertion of carbon monoxide in hydroxymethyl phenol **19** through a Pd-catalyzedcatalyzed carbonylative reaction (Figure 2). Note that the spectroscopic data for synthetical **5** are in agreement with those described for natural pterolobirin H (**5**) isolated from Pterolobium macropterum [16].

It is important to point out that the stereochemistry of the C-5 and C-10 in all synthesized compounds is preserved. The process used for constructing the aromatic ring of the target compounds, which is based on the Diels–Alder cycloaddition reaction, does not modify the stereochemistry of said asymmetric carbons.

The subsequent reduction in lactone **5** with DIBAL-H afforded lactol **6**, whose spectroscopic data are in agreement with those described for pterolobirin G (**6**) [16]. Dehydration of **6** with Amberlyst A-15 afforded (5a)-vouacapane-8(14),9(11)-diene (**7**) (Figure 3).

Next, some 14-nor-cassane compounds, analogs of the previously selected compounds, were synthesized from (-)-ferruginol (**12**). First, the synthesis of 14-demethylsalicylaldehyde **23** was investigated (see Figure 4). To this end, the phenolic group in **12** was protected by methyl ether and the formyl group was introduced at the C-13 via ipso-substitution reaction of the isopropyl group [7]. Deprotection of the methyl group afforded 14-desmethyl salicylaldehyde **23** with an excellent yield (95%). 

Following a similar sequence of reactions used for the preparation of pterolobirin G (**6**) [16], 14-desmethyl lactone **25** was obtained. (Figure 5).

### 2.2. Biological Assays

#### 2.2.1. Cytotoxicity on RAW 264.7 Cell Line

To perform the anti-inflammatory tests and establish the values of the sub-cytotoxic concentrations of the newly synthetized diterpenoids and the 14-desmethyl analogs (**5**–**7**, **14**–**20** and **23**–**25**), cytotoxic effects of these diterpenes were analyzed on RAW 264.7 monocyte/macrophage murine cells. Cell viability was determined using the MTT (3-(4,5-dimethyl thiazol-2-yl)-2,5-diphenyltetrazolium bromide) assay, with increasing concentration levels of each compound (0–100 μg/mL). MTT is transformed to formazan in cells with metabolism capacity (viable cells), as the number of viable cells is proportional to formazan concentration; this assay was used to determine the cytotoxicity of the compounds. The compound concentrations required for 20%, 50%, and 80% cell viability inhibition (IC_20_, IC_50_, and IC_80_) after 72 h of incubation were determined by the absorption of a formazan dye to analyze the complete range of cytotoxicity on RAW 264.7 cells (Table 1). Among the tested compounds, hydroxyester **17** was the most cytotoxic, with an IC_50_ value of 7.49 ± 0.86 μg/mL, followed by the salicylaldehyde **20**, IC_50_ 8.34 ± 2.51 μg/mL, hydroxyester **18** with 13.11 ± 1.51 μg/mL and lactol (**6**), IC_50_ value 14.99 ± 0.32 μg/mL. For the rest of the compounds, the range of their broad-spectrum cytotoxic activities inferred by IC_50_ data was moderate-to-weak: between 24.67 ± 0.04 and 74.65 ± 0.17 μg/mL. (Table 1). 

Based on these results, we determined the sub-cytotoxic concentrations corresponding to IC_50_, ¾ IC_50_, ½ IC_50_, and ¼ IC_50_, which were used in the next assays to ensure that the anti-inflammatory activity of our compounds was due to their anti-inflammatory process and not to their intrinsic cytotoxicity. The study of the NO release inhibition of all thirteen products was realized.

#### 2.2.2. Effects on Inhibition of NO Production

The activation of the inflammatory process induces nitric oxide, NO, released as a second messenger. NO production can be used as the main indicator to assess inflammatory activity, which can be measured using nitrite (NO^2–^) quantification (proportional to the released NO), determined through the Griess reaction. Here, the selected compounds (**5**–**7**, **14**–**20** and (**23**–**25**) were tested for their ability to inhibit NO production in LPS-stimulated RAW 264.7 murine macrophage cells (Figure 2). These macrophages were activated with LPS during the 24 h after the addition of the synthesized compounds. The sub-cytotoxic concentrations ¾ IC_50_, ½ IC_50_, and ¼ IC_50_ were used, and nitrite concentrations were determined in a cell culture medium at 72 h of incubation (Figure 2). The results showed that all tested compounds exhibited crucial inhibition of NO release. Inhibition percentages were calculated with respect to the increase between positive control (only LPS treated control cells) and negative control (untreated control cells). 

All compounds were able to inhibit the NO release in a dose-dependent manner. Thus, compound **16** reached a higher anti-inflammatory effect, with 100% of NO-inhibition at the three concentrations assayed ¼ IC_50_, ½ IC_50_ and ¾ IC_50_. Eight out of the thirteen selected cassane diterpenes (**5**–**7**, **14**–**16**, **19**–**20**), displayed a strong NO inhibition between 78% and 100% at ¾ IC_50_ concentration. The inhibition achieved by compounds **17** and **18** was lower: 30.52% and 11.06%, respectively, at ¾ IC_50_ concentration (Figure 2A). Concerning the desmethylated analogs (**23**–**25**) (see Figure 2C), compounds **23** and **25** showed the highest anti-inflammatory effect at ¾IC_50_ concentration, with a 78.48% and 72.24% of NO inhibition, respectively. However, at the same sub-cytotoxic concentration, product **24** did not cause any inhibition of NO release (Figure 2).

Therefore, our results are in agreement with the findings previously reported in the bibliography; for example, the natural compound **7** showed 34.5% inhibition of NO release at 10 μmol/L after 3 h of treatment in RAW 264.7 activated with 100 ng/mL of LPS [11]. Recently, furan **7** and pterolobirin G (**6**) were isolated from the roots of *Pterolobium macropterum* and analyzed for their anti-inflammatory potential against NO production in LPS-induced J774.A1 macrophage cells. The authors have found that the natural furan **7** and the natural pterolobirin G (**6**) did not show any significant inhibition of NO production at 50 μM in this cell type [16].

To complete our anti-inflammatory study, we calculated the concentration that reduces the production of NO (IC_50 NO_) to 50% after 72 h of cells incubation with the assayed compounds. Compound **20** showed the highest effectiveness (IC_50 NO_ = 2.98 ± 0.04 μg/mL), followed by its demethylated analog **23** (IC_50 NO_ = 3.22 ± 0.05 μg/mL). Additionally, compounds **6**, **15**, **16**, and **17** exhibited important NO inhibition, with IC_50 NO_ values between 4.01 μg/mL and 7.98 ± 0.1 μg/mL (see Table 2).

The methyl, vinyl, or carboxymethyl functional group linked at C-14 position has been described as a common structural characteristic in most of the anti-inflammatory cassane-type diterpenoids [21]. This fact is consistent with our results of NO-release inhibition after 72 h, since two out of the synthesized cassane diterpenoids (**5** and **20**) with a methyl group at C-14, exhibited more NO inhibition than their 14-desmethyl analogs **23** and **25** (Table 2).

#### 2.2.3. Effects on RAW 264.7 Cell Cycle Arrest and Distribution

Two compounds with the highest anti-inflammatory effect, the salicylaldehyde **20** (IC5_0 NO_ = 2.98 ± 0.04 μg/mL) and the dienone **16** (IC_50 NO_ = 5.71 ± 0.14 μg/mL), were selected to determine the changes in the cell cycle in RAW 264.7 cells, activated by LPS after 72 h of incubation at different sub-cytotoxic concentrations (¼ IC_50_, ½ IC_50_, and ¾ IC_50_). The cell-cycle distribution was analyzed by flow cytometry with propidium iodide (IP) staining (Figure 3). Significant growth arrest of the cell cycle in LPS-induced RAW 264.7 cells produced 100% detention in the G0/G1 phase (positive control) compared to the negative control (49.6%). After incubation with compound **20**, the percentage of cells found in the G0/G1 phase was 29%, 32%, and 36%, respectively, at the assayed concentrations. Similar results were found for compound **16** (35%, 38%, and 39%) after 72 h of incubation at the same concentrations. This decrease was accompanied by the consequent increase in the number of cells in the S phase: 70%, 68%, and 56%, respectively, for compound **20**, and 63%, 61%, and 59% for compound **16**. The changes in the G2/M phase were not significant. This recovery of the cell cycle with respect to control cells (only LPS treated) could be a consequence of the anti-inflammatory effect produced by the tested compounds, which increase cell division, rescuing RAW 264.7 cells from LPS-induced arrest during their monocyte/macrophage differentiation process.

#### 2.2.4. Cell Cytotoxicity Assay and SAR

Studies were conducted on the cytotoxic effects of the ten synthetic diterpenic cassanes **5**–**7** and **14**–**20** and of the three demethylated analogs **23**–**25** on B16-F10, HT29, and HepG2 cancer cell lines. These studies were performed using the MTT assay, based on formazan dye formation and expressed as a percentage with respect to untreated control cells (100% of cell metabolism). Cells were incubated with increasing concentrations of each compound (0–100 μg/mL) for 72 h. Then, the concentrations of the compounds required for a descent of 50% in the cell viability (IC_50_) were determined. (Table 3). 

All cassane diterpenes **5**–**7** and 14–20 displayed cytotoxicity in the assayed conditions, with IC_50_ data between 2.38 ± 0.39 and 74.33 ± 2.54 μg/mL, showing six compounds (**6**, **14**, **16**, **17**, **18**, and **20**), with an IC_50_ value < 10 μg/mL in B16-F10 murine melanoma cells (IC_50_ values between 2.4 μg/mL to 9.2 μg/mL). Considering the three cell lines, the most active compounds were salicylaldehyde **20** with an IC_50_ value of 2.38 ± 0.39 μg/mL in B16-F10 line and 3.54 ± 0.19 μg/mL in HT29 cells, followed by compound **17** with IC_50_ values 5.96 ± 0.55 in HT29 cells, 8.15 ± 0.10 in HepG2 cells, and 5.96 ± 0.55 μg/mL in B16F10 cell line. 

Moreover, the obtained IC_50_ data of the 14-desmethyl cassane analogs were in the range of 4.33–77.97 μg/mL. To continue divulging the role played by the C-14 methyl group in cassane diterpenes in the antiproliferative effect, we compared the IC_50_ values of salicylaldehyde **20** and lactone **5** with those of 14-desmethyl analogs **23** and **25**, respectively. (Entries 1, 13: and 10, 11: Table 3). As the highest values of cytotoxicity were found for compounds **6** and **20** in a HT29 colon cancer cell, these compounds and cell line were selected for the following assays.

It is observed that the presence of the C-14 methyl group is unimportant for the cytotoxic effect of the salicylaldehyde **20** in all cell lines and the lactone **5**, only in HT29 cells. From this initial *SAR* data analysis, it could be deduced that the methyl group at the C-14 positions of this family of compounds is not always crucial for their cytotoxic activity. This conjecture agrees with the results of a study by Pitsinos et al. [5,6].

#### 2.2.5. Effects on HT29 Cell Cycle Arrest and Distribution

Flow cytometry was used to measure DNA ploidy and to determine possible alterations in cell cycle profiles. The distribution of cells in different cell-cycle phases was analyzed after 72 h of treatment by the incorporation of propidium iodide (PI), whose fluorescence is directly proportional to the amount of DNA. HT29 colon adenocarcinoma cells were treated with the highest cytotoxic compounds **6** and **20** at IC_50_ and IC_80_ concentrations. DNA histogram analysis showed that both of them dose-dependently produced a significant cell-cycle arrest in the G0/G1 phase of the cycle (Figure 4) At the IC_50_ concentration, salicylaldehyde **20** increased this population with 6%, and with 9% at an IC_80_ concentration. Lactol **6** produced a major effect with an increase of 13.6%, and 29% at an IC_80_ concentration, with respect to the untreated cells control. These G0/G1 increases were accompanied by the concomitant decreases in the S phase cell percentages (by 6% and 10.7%) at IC_50_ and IC_80_ concentration, respectively, for compound **20**, and by 17.3% and 30.36% for compound **6** at IC_50_ and IC_80_ concentrations, respectively. Changes in the G2/M phase were less significant. Although changes in the cell cycle were observed, they were of a lower magnitude and insignificant for compound **20**. Therefore, those changes could be produced by the apoptotic process activation and not by cytostatic or differentiation process induction. Nevertheless, for compound **6**, the values reached 30% at IC_80_ concentrations, which could be related to the activation of some of these processes. Further assays will be necessary to asseverate this point.

#### 2.2.6. Characterization of Apoptotic Effects by Flow Cytometry with Annexin-V

Most currently used anticancer drugs trigger the apoptosis process as a key target and induce it in cancer cells. Phosphatidylserine (PS) externalization is a typical characteristic of early phase apoptosis [22]. In this context, we further explored whether cytotoxic and cytostatic effects of salicylaldehyde **20** and lactol **6** on HT-29 cells were related to the induction of apoptosis, using the Annexin V-FITC/PI double staining by flow-activated cell sorter (FACS) cytometry analysis. As illustrated in Figure 5, this double-staining method differentiated four cell populations: normal cells (annexin V− and PI−), early apoptotic cells (annexin V+ and PI−), late apoptotic cells (annexin V+ and PI+) and necrotic cells (annexin V− and PI+). These assays on the HT29 cell line were measured 72 h after treatment with the selected cassane diterpenes at their corresponding IC_50_ and IC_80_ concentrations. Both compounds demonstrated apoptotic effects on HT29 cells, with total apoptosis (early apoptotic together with late apoptotic cells) rates ranging from 18.3% to 86.8%, with necrosis rates that did not exceed 5%, concerning control cells. 

These percentages of total apoptosis reached very high levels up to 76.25% and 86.8% at IC_50_ and IC_80_ concentrations, respectively, in response to salicylaldehyde 20, consisting of (27.57% early apoptosis, 49.97% late apoptosis) at IC_50_ concentration and (41.40% early apoptosis, 45.40% late apoptosis) at IC_80_ concentration. These data agree with the cell cycle arrest at the G0/G1 phase caused by salicylaldehyde **20** at IC_50_ (63.6%) and IC_80_ (66.6%) in the HT29 cell line, which allows us to suppose that the cytotoxic effect of compound **20** occurs through an apoptotic response activation. The treatment of HT29 cells with lactol **6** has also been shown to induce apoptosis, in a concentration-dependent manner. The lowest percentage of total apoptosis was observed at an IC_50_ concentration, of 18.30%, while the highest percentage of apoptosis exerted by compound **6** was obtained at the IC_80_ concentration, close to 82% (48.83% early apoptosis plus 23 % late apoptosis). These results concord with the cell cycle arrest data since the cell percentages in the G0/G1 phase also changed in a concentration-dependent manner. Finally, the percentages of the necrotic populations were unremarkable in response to treatment by both tested cassane diterpenes. Various investigations have established a connection between different signaling pathways that induce cell apoptosis and the antitumor effects of cassane diterpenoids [3,23,24,25] (Figure 5).

#### 2.2.7. Effects on Changes in Mitochondrial-Membrane Potential

The apoptotic effects of anticancer agents can occur through the activation of two major apoptotic pathways: intrinsic and extrinsic apoptotic pathways, Thus, the study of the loss of the mitochondrial membrane potential (MMP) of tumor cells can provide information on the activation mechanism and the apoptotic pathway induced in response to anticancer compounds. When the mitochondria undergo disruption and are involved in the loss of MMP, the activation is then intrinsic (mitochondrial). Apoptosis induction without MMP alterations may suggest the activation of the extrinsic apoptotic pathway. Changes in MMP were evaluated using flow-cytometry double staining with Rh123 and PI in colon adenocarcinoma HT29 cells after treatment with compounds **20** and **6** for 72 h at IC_50_ and IC_80_ concentrations. Rh123 is a membrane-permeable, a fluorescent cationic dye that is taken up selectively by the mitochondria and its fluorescence is proportional to MMP. The results showed positive Rh123 staining in HT29 cells for both lactol **6** and salicylaldehyde **20** at IC_50_ and IC_80_ concentrations, so apoptosis induction by these compounds did not produce any changes in MMP, suggesting that the apoptosis was triggered by the extrinsic pathway activation (Figure 6).

Several studies on the apoptotic mechanisms behind the cassane diterpenoids anticancer activities showed that some of these diterpenoids induce apoptosis through different mechanisms. For instance, phanginin D triggers apoptosis by activating caspase-3, phanginin R induces apoptosis by enhancing the (PARP) activity, and procaspase-3 cleavage also increases the ratio of Bax/Bcl-2 and the p53 protein expression [23,24]. Additionally, caesalpin G induces apoptosis via another apoptotic mechanism, promoting ER stress and suppressing the Wnt/β-catenin signaling pathway [3,25,26].

## 3. Experimental Section

### 3.1. Synthesis

Experimental procedures, product characterizations, and ^1^H and ^13^C NMR spectra are included in the Appendix A.

### 3.2. Biological Experimental Procedures

#### 3.2.1. Test Compounds

Compounds **5**–**6**, **14**–**20**, and **23**–**25** were dissolved at 5 mg/mL in DMSO. A stock solution was stored at −20 °C and before the treatment this solution was diluted in a cell-culture medium to adequate concentrations for each experiment.

#### 3.2.2. Cell Cytotoxicity Assay

The human colorectal adenocarcinoma cell line HT29 (ECACC no. 9172201; ATCC no. HTB-38), human hepatocarcinome cell-line HepG2 (ECACC no. 85011430), mouse melanoma cells B16-F10 (ATCC no. CRL-6475), and murine monocyte/macrophage-like RAW 264.7 cell line (ATCC no, TIB-71) were purchased from the cell bank of the University of Granada, Spain. The cells were cultured in DMEM (Dulbecco’s Modified Eagle’s medium) supplemented with 2 mM glutamine, 10% heat-inactivated FCS (Fetal Calf Serum), 10,000 units/mL of penicillin, and 10 mg/mL of streptomycin, (for all cancer cell lines), 50 μg/mL of gentamicin (only for RAW 264.7 cell line). Cells were incubated at 37 °C, in an atmosphere of 5% CO_2_ and 95% humidity. In all experiments, subconfluent monolayer cells were used. The culture media were changed every 48 h and the confluent cultures were separated with a trypsin solution (0.25%-EDTA).

The effect of compounds on the cell viability was determined by the MTT method [27] (Sigma, St. Louis, MO, USA)**.** The cytotoxicity of the compounds was detected by measuring the absorbance of MTT dye staining of cells with metabolism capacity. The cells were grown in 96-well plates to a volume of 100 μL at 6.0 × 10^3^ cells/mL for HT29 and RAW 264.7 cell lines at 5.0 × 10^3^ cells/mL for B16-F10 cells and 15.0 × 10^3^ cells/mL for the HepG2 cell line and were incubated with the different products (0–100 μg/mL). Finally, after 72 h, 100 μL of MTT solution (0.5 mg/mL) in 50% of PBS and 50% of the medium was added to each well. After 1.5 h of incubation, formazan was re-suspended in 100 μL of DMSO, and each concentration was tested in triplicate. Relative cell viability, concerning untreated control cells, was measured by absorbance at 570 nm on an ELISA plate reader (Tecan Sunrise MR20–301, TECAN, Grödig, Austria). Compounds with low IC_50_ values (**16**, **20**, and **6**) were selected for several cytometry assays, such as apoptosis, cell cycle, and mitochondrial membrane potential determination.

#### 3.2.3. Measurement of Nitric Oxide Concentration

RAW 264.7 cells were plated at 6 × 10^4^ cells/well in 24-well cell culture plates, supplemented with 10 μg/mL of LPS. After 24 h, cells were incubated for 24 h with compounds (**5**–**7**, **14**–**16** and **19**–**20**) at ¾ IC_50_, ½ IC_50_, and ¼ IC_50_ concentrations. Griess reaction [28] was performed by taking 150 μL of supernatant test samples or sodium nitrite standard (0–120 μM) and mixed with 25 μL of Griess reagent A (0.1% *n*-(1-naphthyl) ethylenediamine dihydrochloride) and 25 μL of Griess reagent B (1% sulfanilamide in 5% of phosphoric acid) in a 96-well plate. After 15 min of incubation at room temperature, the absorbance was measured at 540 nm in an ELISA plate reader (Tecan Sunrise MR20–301, TECAN, Grödig, Austria). The absorbance was referred to nitrite standard curve to determine the concentration of nitrite in the supernatant of each experimental sample. The percentage of NO production was determined, assigning 100% at the increase between negative control (untreated cells) and positive control (cells only treated with 10 μg/mL of LPS).

#### 3.2.4. RAW264.7 Cell Cycle

DNA content is directly proportional to the PI fluorescence; it allows us to determine the percentage of cells in each cell cycle phase. Using a fluorescence-activated cell sorter (FACS) at 488 nm in an Epics XL flow cytometer (Coulter Corporation, Hialeah, FL, USA), cell subpopulations could be visualized with different DNA contents. For this assay, 12 × 10^4^ RAW264.7 murine macrophage/monocyte cells stimulated with LPS were plated in 24-well plates with 1.5 mL of medium and incubated with the compounds under study for 24 h at ¾ IC_50_ and ½ IC_50_ concentrations. By treating cells only with LPS, the positive control was performed. However, the negative control was prepared, exposing cells to the tested compounds without LPS stimulation. After treatment, cells were washed twice with PBS and trypsine solution, then resuspended in TBS 1X (10 Mm Tris, 150 Mm NaCl) and then Vindelov buffer (100 mM Tris, 100 Mm NaCl, 10 mg/mL Rnasa, 1 mg/mL PI, pH 8) was added. The samples were placed for 15 min on ice. Immediately before FACS analysis, cells were stained with 20 μL of 1 mg/mL PI solution. Data were analyzed to determine the percentage of cells in each cell cycle phase (G0/G1, S, and G2/M).

#### 3.2.5. HT29 Cell Cycle

The method was performed by flow cytometry after propidium iodide staining (PI). HT29 cells were plated in 24-well plates at a density of 5 × 10^4^ cells/well, with 1.5 mL of medium and incubated, and after 24 h were treated or not (control) with IC_50_ concentrations of the compounds **6** and **20** for 48 h. The cells were then washed twice with PBS, trypsinized and resuspended in 1 × TBS (10 mM Tris and 150 mM NaCl), and after that Vindelov buffer (100 mM Tris, 100 mM NaCl, 10 mg/mL Rnase, and 1 mg/mL PI, at pH 8) was added. After that, cells were stored on ice and were stained with 20 µL of 1 mg/mL PI solution just before measurement. Approximately 10 × 10^3^ cells were analysed in each experiment. The experiments were performed three times with two replicates per assay. Finally, the samples were analysed using a flow cytometer and the number of cells in each stage of the cell cycle was estimated by fluorescence-associated cell sorting (FACS) at 488 nm in an Epics XL flow cytometer (Coulter Corporation, Hialeah, FL, USA).

#### 3.2.6. Annexin V-FICT/Propidium Iodide Flow-Cytometry Analysis

Annexin V and PI double staining was detected by flow cytometry, to confirm the pro-apoptotic effect of compounds **6** and **20**. As such wise, apoptosis was assessed by flow cytometry using a FACScan (fluorescence-activated cell sorter) flow cytometer (Coulter Corporation, Hialeah, FL, USA). For this assay, 5 × 10^4^ HT29 cells were plated in 24-well plates with 1.5 mL of medium and incubated for 24 h. Subsequently, the cells were treated with the selected compounds in triplicate for 24, 48, and 72 h at their corresponding IC_50_ and IC_80_ concentrations. The cells were collected and resuspended in a binding buffer (10 mM HEPES/NaOH, pH 7.4, 140 mM NaCl, 2.5 mM CaCl_2_). At room temperature in darkness, Annexin V-FITC conjugate (1 μg/mL) was then added and incubated for 15 min. Cells were stained with 5 μL of 1 mg/mL PI solution, just before the analysis. In each experiment, approximately 10 × 10^3^ cells were analyzed, and the experiment was duplicated twice.

#### 3.2.7. Flow-Cytometry Analysis of the Mitochondrial Membrane Potential

The electrochemical gradient across the mitochondrial membrane was studied by analytical flow cytometry using dihydrorhodamine (DHR). For this assay, 5 × 10^4^ HT29 cells were plated in 24-well plates, incubated for 24 h and treated with compounds 6 and 20 for 48 h at their corresponding IC_50_ and IC_80_ concentrations. The medium renewal was achieved by adding fresh medium with DHR to a final concentration of 5 mg/mL. Cells were incubated for 1 h at 37 °C in an atmosphere of 5% CO_2_ and 95% humidity and subsequently washed and resuspended in PBS with 5 µg/mL of PI. The fluorescence intensity was measured using a FACScan flow cytometer (fluorescence-activated cell sorter). The experiments were performed three times with two replicates per assay.

#### 3.2.8. Statistical Analysis

Experimental cytotoxicity data were fitted to a sigmoidal function (y = ymax/ (x/a)^−b^) by non-linear regression. IC_20_, IC_50_, and IC_80_ values (concentrations that cause 80%, 50%, and 20% of cell cytotoxicity, respectively) were obtained by interpolation. These analyses were performed using SigmaPlot^®^ 12.5 software. Similar analyses were performed to obtain the IC_50_ of NO production (IC_50 NO_). All shown data are representative of at least two independent experiments performed in triplicate. All quantitative data were summarized as the means ± standard deviation (SD).

More information about experimental procedures can be found in the Appendix A. 

## 4. Conclusions

In conclusion, the high cytotoxic potential and high apoptosis levels attained by our synthetic cassane-type diterpenoids show that they could be used as promising anticancer/anti-inflammatory drugs. Further studies will be necessary to strengthen this point, to elucidate the cellular and molecular elements involved in their effects, and to evaluate the activity/toxicity levels in preclinical models. Additionally, more synthetic strategies should be processed to disclose an expedient entry and to identify more potent anticancer/anti-inflammatory agents based on these novel scaffolds. Suitably, new synthesis, and biological evaluation of additional related structures and cassane analogs are currently underway in our laboratories.

## Data Availability

Data is contained within the article or Appendix A.

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
