# Peer review of "Synthesis and Biological Evaluation of Cassane Diterpene (5α)-Vuacapane-8(14), 9(11)-Diene and of Some Related Compounds"

_molecules, 2022, doi:10.3390/molecules27175705_

Round 1

Reviewer 1 Report

In the manuscript entitled “Synthesis and biological activities of pterolobirin H, pterolobirin G, (5α)-vouacapane-8(14), 9(11)-diene, and 14-desmethyl analogs” the authors synthetized 10 cassane diterpenes and establish their anti-cancer and anti-inflammatory properties. Additionally, 14-desmethyl analogs were also synthetized. The study seems quite interesting but, there is a need to address few queries prior to its acceptance.

1.     The title must be reconsidered because it is not inferred why the proposed compounds are synthesized, what is the point of synthesizing them? In addition, it is indicated that 14-desmethyl analogues are synthesized but it is not clear what the analogues are. In short, the title is neither clear nor correctly conveys the work done.

2.     Keyworks seems poor, considerer included more specific keywords.

3.     Tow authors have an “*” indicating that both are corresponding authors, however only one email ([email protected]) is given.

4.     4. In Scheme 3 legend, why some numbers are in red?

5.     The quality of all figures is very poor. Considerer to generate figures/plots/charts using a different software and generate high quality figures. Always maintain the size and font. Please use Molecules guide for authors to prepare the figures correctly.

6.     The authors stated that all compounds are prepared starting from compound (8), and assume that the stereochemistry is maintained until the end, however there is no evidence that the stereochemistry is indeed maintained in the derivatives. Please expand this discussion to include evidence of the stereochemistry of the compounds.

Unfortunately, I was unable to access the supplementary material from the journal's review system, so the spectroscopic data of the compounds could not be analyzed to verify the information in the paper.

Author Response

1) The title must be reconsidered because it is not inferred why the proposed compounds are synthesized, what is the point of synthesizing them? In addition, it is indicated that 14-desmethyl analogues are synthesized but it is not clear what the analogues are. In short, the title is neither clear nor correctly conveys the work done.

Following the suggestion of the reviewer, the title has been changed to “Synthesis and Biological Evaluation of Cassane Diterpene (5α)-vouacapane-8(14), 9(11)-diene and of somes Related Compounds”

2) Keyworks seems poor, considerer included more specific keywords.

We have followed the recommendation of the reviewer. The new Keywords are “natural products, semisynthesis, cassane diterpenoids; anti-inflammatory activity, anti-cancer activity

3) Tow authors have an “*” indicating that both are corresponding authors, however only one email ([email protected]) is given.

e-mail ([email protected]) was included

4) In Scheme 3 legend, why some numbers are in red? Corrected

5) The quality of all figures is very poor. Considerer to generate figures/plots/charts using a different software and generate high quality figures. Always maintain the size and font. Please use Molecules guide for authors to prepare the figures correctly.

According to reviewer’s suggestion, new figures with high quality (450 dpi) we have included

6) The authors stated that all compounds are prepared starting from compound (8), and assume that the stereochemistry is maintained until the end, however there is no evidence that the stereochemistry is indeed maintained in the derivatives. Please expand this discussion to include evidence of the stereochemistry of the compounds.

 We have included after the scheme 2 the sentence “It is important to point out that the stereochemistry of the C-5 and C-10, in all synthesized compounds, is preserved. The process used for constructing the aromatic ring of the target compounds, which is based on the Diels-Alder cycloaddition reaction, does not modify the stereochemistry of said asymmetric carbons”

Reviewer 2 Report

The reviewer is mainly concerned with the section of synthetic methods.

The present manuscript describes syntheses of a series of ten cassane diterpenoids and the evaluation of anti-cancer, anti-inflammatory bioactivities.  Extensive studies of SARs of these fourteen compounds including natural products, accessible synthetic analogues, and synthetic intermediates, are investigated.  These studies were implemented by many members of R-Zurita and Chahboun groups.  Characterization of these compounds were well-performed.

More detailed synthetic process of 17 and 20 in Scheme 2 should be described in the text.

Regarding SARs, salicyaldehyde type compounds 20 and 23 exhibited significant bioactivities among tested series.  Is the author able to provide speculative reasons for the different mode of action based on the aldehyde structure?

On the whole, the reviewer recommends the publication in Molecules. 

<Comments and suggestions>

1.      p. 3, line 89; easly → easily

2.      p. 4, line 92; the same → the similar

3.      p. 4, line 101; 20 19

4.      p. 5, line 107; What is 23??

5.      p. 5, line 108; give → afforded

6.      p. 5, line 113; prepared → synthesized

7.      p. 5, line 113; ferruginol (12) → Is ferruginol (12) available? Please address this issue.

8.      p. 5, line 113; #######aldehyde is prepared. → the synthesis of #######aldehyde (23) was investigated.    

9.      p. 5, line 113; To do this → To this end

10.   p. 5, line 114; group of ferruginol (12) → group in 12

11.   p. 5, line 114; was protected as a methyl ether group → was protected by methyl ether

12.   p. 5, line 115; the number of “C-13” should be provided for compounds 21 and 22 in Scheme 4.  In addition, the citation of this transformation method should be provided.

13.   p. 5, line 116; the deprotection of phenolic group leads to → the deprotection of methyl group afforded

14.   p. 5, line 117; “(Scheme 4)” should be move after the end of the sentence in line 113.

15.   p. 5, line 120; the same → the similar

Author Response

We thank all comments and suggestions made by the reviewer 

Reviewer 3 Report

·        English grammar and style should be corrected.

·        All abbreviations should be defined when used for the first time.

·        Abstract is chaotic and sounds as not very interesting. Should be rewritten. It should be more structured: contain a brief introduction, purpose, methods (almost non- existent now), and a description of the results and brief conclusions.

·        The aim of the study should be clearly defined.

·        Authors could emphasise the novelty of the study in the Introduction.

·        ‘Caesalpinia’ should be written in italics.

·        MTT is not cell proliferation assay (it measures cells metabolism). It should be corrected in the whole manuscript. It is also not a test for anticancer activity testing.

·        Why MTT was done for all cell lines, while other test only for HT-29? Justify in the text please.

·        Is the method of synthesizing these compounds on a commercial scale profitable? Discuss it please.

Author Response

(The authors gave the same response as above.)

Round 2

Reviewer 1 Report

The authors addressed all the comments.